# The Status of Oxidative Stress in Patients with Alcohol Dependence: A Meta-Analysis

**DOI:** 10.3390/antiox11101919

**Published:** 2022-09-28

**Authors:** Mi Yang, Xiaofei Zhou, Xi Tan, Xincheng Huang, Lu Yuan, Zipeng Zhang, Yan Yang, Min Xu, Ying Wan, Zezhi Li

**Affiliations:** 1The Fourth People’s Hospital of Chengdu, Chengdu 610036, China; 2The Clinical Hospital of Chengdu Brain Science Institute, MOE Key Lab for Neuroinformation, University of Electronic Science and Technology of China, Chengdu 611731, China; 3School of Life Science and Technology, University of Electronic Science and Technology of China, Chengdu 611731, China; 4The Affiliated Brain Hospital of Guangzhou Medical University, Guangzhou 510370, China; 5Guangdong Engineering Technology Research Center for Translational Medicine of Mental Disorders, Guangzhou 510370, China

**Keywords:** alcohol dependence, oxidative stress, antioxidant, superoxide dismutase, glutathione peroxidase

## Abstract

Alcohol-induced oxidative stress (OS) plays a pivotal role in the pathophysiology of alcohol dependence (AD). This meta-analysis was aimed at investigating the changes in the levels of OS biomarkers in AD patients. We included relevant literature published before 1 April 2022, from the PubMed, Web of Science, and EBSCO databases following PRISMA guidelines. Finally, 15 eligible articles were enrolled in this meta-analysis, including 860 patients and 849 controls. Compared with healthy controls, AD patients had lower activities of superoxide dismutase (SOD) and glutathione peroxidase (GPx) enzymes, and lower levels of albumin, while levels of malondialdehyde (MDA), vitamin B12, homocysteine, and bilirubin were significantly increased in serum/plasma samples of AD subjects (all *p* < 0.05). In male patients, the activities of SOD and GPx were increased in serum/plasma but decreased in erythrocytes (all *p* < 0.05). The opposite trends in the level of SOD and GPx activities in serum/plasma and erythrocytes of male patients could be used as the biomarker of alcohol-induced OS injury, and the synergistic changes of MDA, vitamin B12, albumin, bilirubin, and homocysteine levels should also be considered.

## 1. Introduction

Alcohol dependence (AD) or alcoholism is a complex and serious psychiatric disorder that can lead to perturbations in daily physical, psychological, and social functions [1]. Currently, AD accounts for a prevalence rate of about 2.6% in the general population [2,3,4]. The lack of effective prevention strategies, treatments, and rehabilitation programs are the major contributing factors to the increasing global health burden of AD [5]. AD pathogenesis is highly complex and involves multifaceted etiological factors, including altered neuroplasticity, neuropsychiatric disorders, disoriented socio-environmental interactions, and genetic inheritance.

The imbalance between the rate of generation of toxic free radicals such as reactive oxygen species (ROS) and reactive nitrogen species (RNS) and compromised or insufficient antioxidant defense response can cause chronic oxidative injury to the human body. Although low levels of ROS and RNS play a vital role as secondary signal transducers and gene activators under normal physiological conditions [6,7], persistently accumulating non-neutralized excess free radicals lead to a broad spectrum of oxidative damages to almost all types of major macromolecules, such as DNA, RNA, proteins, and lipids, thereby inducing a number of chronic neuropsychiatric and neurodegenerative diseases [8], including mild cognitive impairment (MCI), dementia [9], schizophrenia [10], and Parkinson’s disease (PD) [11]. Normally, the human body, especially the brain, possesses excellent antioxidant defense machinery to prevent damages caused by free radical toxicities. However, this preventive mechanism can be compromised under diseased or abnormal health conditions [12,13,14].

In the brain, ethanol is oxidized to acetaldehyde through the action of catalases (CATs) [15,16], cytochrome P450 enzymes (CYP2E1) [17], and alcohol dehydrogenase (ADH) [18,19]. Notably, CAT and CYP2E1 play major roles in catalyzing the biochemical conversion of ethanol to acetaldehyde [20]. It has been found that the expression of CYP2E1 is induced by long-term drinking habits or AD [21]. The mechanistic functions of CYP2E1 and ADH reportedly produce RNS and ROS, which in turn activate downstream enzymes such as nitric oxide synthase, nicotinamide adenine dinucleotide phosphate (NADP) oxidase, and xanthine oxidase [17]. Furthermore, acetaldehyde consumes reduced glutathione (GSH), perturbing the intracellular redox balance, resulting in oxidative stress (OS) [22]. Therefore, alcohol and its toxic metabolites may be the sole cause of increased cellular burdens of ROS/RNS and other types of highly reactive free radicals and superoxides, leading to the OS injuries to the vital organs of the body [23]. It has been demonstrated that an initial high level of OS can activate the antioxidant defense to scavenge free radicals and prevent lethal free radical chain reactions [24]. Hence, increased activity of antioxidant enzymes, including glutathione peroxidase (GPx), superoxide dismutase (SOD), glutathione reductase (GR), and CAT, is observed in AD patients [25,26]. Moreover, during the process of ROS/RNS neutralization and prevention of systemic free radical chain reactions [27,28], the levels of non-enzymic antioxidants such as vitamin B9 (folate), vitamin B12 [29], GSH [30,31], bilirubin [30], and homocysteine [29] are significantly increased. On the other hand, levels of certain antioxidant enzymes (SOD, CAT, GPX) [30,32] and non-enzymatic molecules (vitamin E/α-tocopherol, albumin, vitamin C/ascorbic acid) [33] remain unchanged or decrease under OS. Ethanol-induced ROS causes oxidative damages in multiple ways, including oxidation of DNA/RNA/protein/lipid molecules, covalent adduct formation between acetaldehyde and membrane lipids [34] initiating lipid peroxidation and malondialdehyde (MDA) production [35,36,37], protein carbonylation [38], and generation of 8-hydroxy-2′-deoxyguanosine (8-OHdG) as a marker of oxidative DNA damage [35]. Therefore, chronic exposure to alcohol or persistent AD can lead to OS-mediated pathological alterations in brain microstructures and the functional connectivity among neuronal circuitry, resulting in cognitive impairment [39,40,41], which may turn into Wernicke’s encephalopathy (WE) [42] or Korsakoff syndrome (KS) [43,44] in the long run. Alcohol-induced OS may also induce pathogenesis of major diseases [45,46,47]. Therefore, due to the inconsistent changes in OS markers and the possible serious health complications of AD patients, it is of utmost importance to delineate the actual level of OS induction in AD subjects.

Here, we conducted a meta-analysis involving the published biomarkers of OS, including levels of antioxidant enzymes, non-enzymatic antioxidants, and OS injury-associated by-products commonly found in the blood of AD patients, to provide an objective reference basis for the clinical prevention and rehabilitation treatment.

## 2. Materials and Methods

The Preferred Reporting Items for Systematic Review and Meta-Analysis (PRISMA) guidelines [48] were followed to perform this meta-analysis (CRD42022341481).

### 2.1. Literature Screening

We searched for relevant full-length articles published before 1 April 2022, in PubMed, Web of Science, and EBSCO databases using the keywords “alcohol dependence/dependent” + “oxidative/oxidant/antioxidative”. The inclusion criteria of eligible articles were as follows: (1) the subjects were diagnosed with AD, according to the Diagnostic and Statistical Manual of Mental Disorders (DSM) or International Classification of Diseases (ICD) criteria at admission; (2) patients’ blood or other samples were collected within 24 h of admission, such that patients’ health status could be assessed before initiating any treatments for AD symptoms, and the study data should include outcome indicators related to OS; (3) healthy control data were included; (4) articles were peer-reviewed and published in the English language; and (5) if the same experimental samples were measured for the same indexes, the research articles with the largest sample sizes or the most detailed data were selected. The exclusion criteria included: (1) studies involving animal and/or in vitro experiments; (2) conference abstracts, books/chapters, reviews, meta-analyses editorials/letters, and other non-research articles; (3) incomplete data, missing controls, or patients were not diagnosed by DSM or ICD; and (4) the authors could not provide explanation or details of incomplete or missing information. The study retrieval and inclusion and exclusion processes are shown in Figure 1.

### 2.2. Quality Assessment

Since the purpose of this study was to analyze differences in OS levels between AD patients and healthy controls, the quality assessment for case-control studies was performed using the Newcastle–Ottawa scale (NOS) (http://www.ohri.ca/programs/clinical_epidemiology/oxford.asp, accessed on 5 May 2022) to evaluate the quality of chosen articles. The NOS is divided into three parts and includes a total of eight items. Each item scores 1 star except the comparable items that can score 2 stars, indicating the comparability between the patients and controls (e.g., age and gender) so that the scale can obtain a maximum of 9 stars. Two clinical researchers independently scored the included literature. In case of any inconsistency in the evaluation of a certain index of an article, the primary investigators first discussed the matter. However, if it remained unresolved, a third researcher was involved for arbitration. The two researchers first received the necessary training. Then, kappa (κ) test showed a strong consistency (κ = 0.87) in the judgment of these two investigators.

### 2.3. Data Extraction

The following information was extracted from the included articles: the first author’s surname, publication year, sample size, patient diagnosis method, demographic indicators (sex, average age, drinking history, alcohol intake), OS indicators (sample type, data mean/standard deviation, unit), if all values were converted into the form of mean and standard deviation [49,50,51].

### 2.4. Data Analysis

The RevMan5.4 (https://training.cochrane.org/online-learning/core-software-cochrane-reviews/revman, accessed on 3 March 2022) software was used for meta-analysis. The *I*^2^ test was used to analyze the heterogeneity of literature that included various OS parameters. The random effect (RE) model was used to assess the large heterogeneity in the index when *I*^2^ > 50%. The fixed effect (FE) model was applied to estimate the minor heterogeneity of the index when *I*^2^ ≤ 50%. When *I*^2^ = 0, the results of the RE and FE models were fully consistent, and we chose the RE model for analysis. Sensitivity analysis measured the contribution of the literature to the combined effect by the one-by-one elimination method. OS index analysis showed variations depending on the sample type such as serum, plasma, red blood cell (RBC), and possibly the male gender. Because the outcome index has different units such as µg/mL, U/g, and U/mL, the differences in OS indexes between AD patients and healthy controls were quantified by standard mean difference (SMD). The combined effect was considered significant when *p* ≤ 0.05.

## 3. Results

### 3.1. Literature Screening

A total of 397 records were retrieved from PubMed (*n* = 50), Web of Science (*n* = 161), and EBSCO (*n* = 186) databases, and 5 references were included. First, 43 repeated records and 11 non-English articles were excluded. Second, 45 in vitro experiments, 83 books or chapters, 79 review articles, 36 correspondences or editorials, and 44 animal experiments were excluded based on the study topic and abstract. Next, we reviewed the remaining 61 full-length articles, further excluding 3 non-research articles, 11 non-AD studies, 15 articles without control data, 9 articles with inappropriate experimental design, 7 articles without patients’ diagnostic criteria, and 1 article for not having the full text available online. The retrieval and exclusion processes are illustrated in Figure 1. Finally, 15 articles were included for meta-analysis [25,26,29,30,31,32,33,35,36,37,38,52,53,54,55], including 860 AD patients and 849 healthy controls (Table 1).

### 3.2. Quality Assessment

General information on the included articles is presented in Table 1. Most studies applied the DSM standard, while only two articles involved ICD. The average age of AD patients was slightly higher (~40 years) compared with that of healthy controls. The average age and sex of patients and healthy controls were matched in most articles. The quality assessment results were expressed on the NOS (Table 2), and the average score of the included literature was >6 points.

### 3.3. Effect Size Estimation

In comparison with the healthy control patients, the results of combined effect size estimation of AD patients are exhibited in Table 3. The enzymatic activities of SOD (RE: SMD = −2.21, *I*^2^ = 97%, 95%CI = [−3.59, −0.82], *Z* = 3.13, *p* = 0.002; Figure 2A) and GPx (RE: SMD = −0.89, *I*^2^ = 91%, 95%CI = [−1.64, −0.15], *Z* = 2.36, *p* = 0.02; Figure 2B) in erythrocytes of AD patients were significantly decreased. In serum/plasma samples, levels of MDA (RE: SMD = 0.89, *I*^2^ = 61%, 95%CI = [0.58, 1.20], *Z* = 5.66, *p* < 0.001; Figure 3A), bilirubin (RE: SMD = 0.50, *I*^2^ = 0%, 95%CI = [0.31, 0.69], *Z* = 5.05, *p* < 0.001; Figure 3C), vitamin B12 (RE: SMD = 0.67, *I*^2^ = 55%, 95%CI = [0.21, 1.12], *Z* = 2.87, *p* = 0.004; Figure 3D), and homocysteine (RE: SMD = 0.98, *I*^2^ = 0%, 95%CI = [0.62, 1.34], *Z* = 2.81, *p* < 0.001; Figure 3E) were significantly increased, while the albumin level (RE: SMD = −1.07, *I*^2^ = 94%, 95%CI = [−1.74, −0.40], *Z* = 3.12, *p* = 0.002; Figure 3B) was significantly decreased. In the serum/plasma samples of male patients, the enzymatic activities of SOD (RE: SMD = 0.30, *I*^2^ = 0%, 95%CI = [0.11, 0.48], *Z* = 3.10, *p =* 0.002; Figure 4A) and GPx (RE: SMD = 0.27, *I*^2^ = 0%, 95%CI = [0.09, 0.46], *Z* = 2.86, *p* = 0.004; Figure 4B) were significantly increased. Likewise, activities of SOD (RE: SMD = −2.21, *I*^2^ = 97%, 95%CI = [−3.59, −0.82], *Z* = 3.13, *p* = 0.002; Figure 4A) and GPx (RE: SMD = −0.89, *I*^2^ = 91%, 95%CI = [−1.64, −0.15], *Z* = 2.36, *p* = 0.02; Figure 4B) were also significantly increased in RBC samples. The serum/plasma levels of MDA (RE: SMD = 1.16, *I*^2^ = 67%, 95%CI = [0.62, 1.71], *Z* = 4.21, *p* < 0.001; Figure 4C) and bilirubin (RE: SMD = 0.40, *I*^2^ = 0%, 95%CI = [0.10, 0.70], *Z* = 2.64, *p* = 0.008; Figure 4D) were significantly higher in male AD subjects. There were no significant differences in other antioxidants between the patients and controls (Appendix A).

### 3.4. Sensitivity and Publication Bias Analysis

In the combined effect size estimation, the quantitative sensitivity analysis method was not used, since there were not enough articles included for statistical assessment. Instead, the one-by-one deletion method was used to perform the analysis. There was no obvious change in the combined effect of all the indicators when removing one at a time. However, in the case of OS biomarkers, the changes were significant when any one parameter was removed. Similarly, due to the limited number of studies included in the funnel plot (Appendix A), we could only qualitatively conclude that there might be some degree of publication bias for some OS indicators. Finally, the risk of bias for each article was assessed, as shown in Appendix A.

## 4. Discussion

Drinking alcohol is an integral part of most national cultures [56,57]. For example, drinking may be regarded as a symbol of friendship and social unity. However, uncontrolled alcohol consumption is one of the top 10 risk factors for death worldwide [1]. However, studies have shown that no level of alcohol consumption is good for health: that is, the safe drinking level is no drinking [58]. Drinking alcohol may have a variety of harmful impacts, such as interpersonal violence [59], suicide and self-harm [60,61], road accident [62,63], drowning [64], work injury [65,66], and serious socio-economic burden as well. Alcohol is a commonly used psychoactive substance. Excessive consumption can cause neuropathological symptoms [67], cardiovascular diseases [68], liver diseases [69], intestinal diseases [70], liver cancers [71], and infectious diseases due to the weakened immune system of the body [72]. The main component of alcoholic drinks is ethanol, which has shown to exert oxidative damages to biological macromolecules via acetaldehyde-DNA/RNA/protein adduct formation, thereby drastically inducing cellular ROS production and systemic OS [14,19,22]. These toxic conditions can then lead to abrupt changes in the levels of antioxidant enzymes and other forms of antioxidant molecules in the body.

SOD plays a major role in antioxidant defense mechanisms [73], especially to protect mitochondrial, cytoplasmic, and peroxisomal membranes [74,75] where it converts superoxide radicals into hydrogen peroxide (H_2_O_2_) molecules, which are then biochemically degraded into water and oxygen by GPx and CAT [76,77], and also regulates the superoxide free radical level in the cell [78,79]. Animal experiments have revealed that the SOD activity in erythrocytes of alcohol-fed rats is significantly lower compared with that of sham-treated animals [80], which is consistent with the significant decrease in SOD activity in erythrocytes of AD patients, especially males, in this study. The phenomena might be explained by the fact that there is a large amount of cytoglobin in erythrocytes, which has a function similar to SOD and can accelerate the disproportionation of superoxide radicals with the catalytic efficiency of SOD [80]. Therefore, cytoglobins may inhibit the enzymatic activity of SOD in RBC by competition due to the greater amount. Additionally, alcohol consumption can lead to the reduction in zinc, which is an essential trace element in the human body as well as an important cofactor of SOD and many critical transcription factors [81]. Alcohol intake also causes deficiency of vitamin D in humans [82], leading to suppressed mRNA expression and enzymatic activity of SOD1 [83]. Other studies have reported that free radicals produced by ethanol metabolism can react with copper and zinc SODs, resulting in their functional inactivation [84]. We noticed that the activity of SOD in serum/plasma samples of AD patients was significantly increased, which might be due to the large pool of toxic ROS and the resulting OS, caused by the degradation of alcohol in the human body. This situation can induce the activation of antioxidant factors, including SOD, to neutralize those free radicals. The hemolysis caused by RBC membrane rupture may also be another reason for the increased SOD or GPx activity in serum/plasma of AD patients. However, it has been shown that the SOD activity in serum/plasma [85], synaptosomes [86], kidney, and liver [86] decreases during the long-term feeding of an alcohol diet to the experimental animals. This inconsistency may indicate that serum/plasma SOD activity in animal experiments may not represent the actual pathobiological scenario in human AD patients, or it could be individualized effects. Furthermore, the activity of SOD in plasma/serum is too low, and the measuring methodology should be essential for precising detection.

GPx commonly refers to the members of glutathione isozyme families that use reduced GSH as an electron donor to break down H_2_O_2_ or organic hydroperoxide into water or corresponding alcohol [87]. The expressions of different subtypes of GPx in different tissues of the human body have their specificities [88,89]. GPx enzymes coordinate with several other signaling molecules to mediate the antioxidant defense processes and inhibit inflammatory responses [88]. GPx plays an important role in promoting the repair of vascular endothelial cells and functionally damaged neurotransmitters following the OS injury and thus helps in delaying cellular aging [90]. Our results showed that the enzymatic activity of GPx in RBCs of AD patients, especially males, was significantly decreased, which was consistent with the findings of a previous study [79]. One possible explanation might be the increased level of acetaldehyde under the OS condition and the resulting inhibition of activities of both GSH and GPx. Other animal experiments have also supported the fact that ethanol exposure can significantly increase the GPx activity of male Wistar rats in the epididymis (21 days) [91] or liver tissues (63 days) [92], which were again consistent with the results of our meta-analysis using human AD patients. Hemolysis-mediated RBC breakdown might increase the GPx activity in serum/plasma of AD subjects. However, the GPx activity in the liver of female mice was significantly decreased after 30 days of ethanol exposure [93] and also in the kidney and liver of male rats [86], suggesting that the effects of alcohol-induced GPx activity may vary in a tissue type and gender-specific manner. Taken together, these factors may partly explain the non-statistically significant changes in plasma GPx activities in AD patients in this study.

We found that the activities of SOD and GPx were enhanced in plasma/serum samples and diminished in erythrocytes of AD patients, while these two enzymes often produced synergistic effects on OS [94]. Therefore, it could be considered in the future as a combined biomarker of OS levels in such patients. Although animal experiments have shown that female and male animals may have different levels of OS during alcohol exposure and that females are more susceptible to alcohol damage [95], due to the lack of sufficient clinical data in female patients, we could not recapitulate that analysis. In the future, a large number of studies are needed to explore whether this observation leads to a different mechanism of OS management in female AD patients than in males.

CAT is a key enzyme in the metabolism of H_2_O_2_ and RNS. In our study, no significant changes in CAT activity were found in serum/plasma or erythrocytes, possibly because CAT could be involved in the oxidative metabolism of ethanol on the one hand [15,16], and in the metabolism of H_2_O_2_ on the other hand, which may have a competitive inter-relationship. In addition, studies have demonstrated that there are no adaptive changes in CAT activity in the myocardium and brain of alcohol-fed rats [96,97], which seems to indicate that alcohol may not affect CAT activity in humans.

MDA, a toxic by-product and one of the biomarkers of OS [98,99], is the most studied product of polyunsaturated fatty acid (PUFA) peroxidation [100]. Our results showed that the most severe lipid-peroxidation-mediated oxidative damages were found in the serum/plasma and erythrocyte membranes of AD patients compared to that in control subjects, which was in agreement with the increased MDA levels observed in the 60-day alcohol-fed albino Wistar rats [79]. In vivo studies have further confirmed that OS-induced lipid peroxidation causes the maximum damage to the erythrocyte membranes of alcohol-exposed rats [79,101], which was in line with our previous results [102,103]. Notably, there could be certain technical artifacts that could influence the above finding. First, hemolysis might become activated during the isolation of erythrocytes from plasma/serum samples, which could then increase the MDA level in the respective samples. Second, membrane phospholipids could undergo rapid peroxidation during the preparation of tissue homogenates, resulting in the overestimation of MDA levels in the downstream analysis. Hence, it is necessary to take preventive measures to avoid any unwanted production of aldehydes in the process of organelle separation [104]. Third, an inappropriate diet (e.g., high protein or fat) can also lead to OS, manifested as an increased level of urinary MDA [105,106]. Moreover, the level of MDA is associated with gender, age [107], vitamin status, and smoking habits [108]. Considering the above possibilities, the MDA level may be considered as the OS biomarker for evaluating the status of erythrocyte membrane damage in AD patients.

Bilirubin is a potent scavenger of ROS and RNS/NO [109,110]. It can modulate the levels of pro-inflammatory cytokines, thereby inhibiting the migration/infiltration of activated immune cells to the lesion sites [111]. Experiments in albino male Wistar rats chronically treated with an alcohol diet for 28 days [112] or 60 days [113], as well as 30 days of alcohol exposure to ICR mice [114], consistently demonstrated a significant increase in the total plasma/serum bilirubin levels, which were in line with our observations in the present study. Bilirubin inhibits the glucuronidation of ethanol via the competitive binding with UDP-glucuronosyltransferase 1A1 [115]. Hence, it is considered an in vivo protective factor [116,117] against the pathological onset of cardiovascular diseases and type 2 diabetes in AD patients. Vitamin B12 (cobalamin) deficiency is a common cause of various neuropsychiatric symptoms [118,119]. Elevated serum B12 levels might be indicative of many serious underlying health complications such as solid tumors, liver cirrhosis, hepatic carcinoma, and chronic renal failure [120,121]. In this study, significantly elevated vitamin B12 levels suggest serious hepatotoxicity in individuals with uncontrolled alcohol consumption, resulting in the dysfunctional vitamin B12 metabolism [121,122], which could be reflected in the dramatic elevation of plasma/serum vitamin B12 levels. Therefore, vitamin B12 could be used as a biomarker to predict the status of liver lesions in this subset of patients [123]. Homocysteine is a sulfur-containing amino acid, and its metabolism is related to the cellular concentrations of folic acid and vitamin B12 [124]. Animal studies using the AD mice model have shown that chronic drinking can significantly increase the level of plasma homocysteine [125,126], which was consistent with our results. Excessive homocysteine can impair various physiological mechanisms, especially the amino acid metabolism pathways [127]. Moreover, it can induce neuronal damage by stimulating the N-methyl-D-aspartate (NMDA) receptor activity and overproduction of toxic free radicals, leading to neurodegenerative conditions, brain atrophy, and withdrawal seizures in susceptible individuals [128]. Serum albumin plays potential roles in anti-inflammatory, antioxidant, anticoagulant, and anti-platelet aggregation mechanisms [129,130]. A mice study exhibited a significant decrease in the serum albumin level after 2 days of alcohol exposure [131]. Similar effects have also been observed in adult Wistar rats following 28 days of alcohol exposure [132]. Taken together, findings from these acute and chronic alcohol exposure studies were consistent with our results, suggesting that reduced albumin levels could be a risk factor for cardiovascular diseases [129], liver diseases [133], and kidney diseases [134]. Thus, changes in albumin levels in AD patients may have certain clinical implications in the diagnosis, treatment, and rehabilitation strategies.

Among the several limitations of this study, the small number of the included articles was a major drawback, which led to the fact that the individual OS-related biomarkers used in our meta-analysis could not be analyzed and corrected for the quantitative publication bias but could only be analyzed from the funnel plot. We speculate that most of the reported indicators may have publication bias, which could be attributed to multiple factors, such as (1) the enrolled studies were published over a long period (1993–2019) and (2) both DSM and ICD scales have undergone significant improvement in their diagnostic standards (DMS-III/IV/V and ICD-10/11). The alcohol abuse and alcohol dependence were combined into alcohol use disorder in DSM-5; therefore, we may exclude some recent research due to the searching strategy. However, there are some differences in status between alcohol abuse and alcohol dependence, which may affect oxidative stress status. First, alcohol abuse (ICD-10-F10.1) patients are those who have suffered physical or mental harm because of alcohol use, but some of these patients may not meet the diagnosis of alcohol dependence (ICD-10-F10.2). Second, alcohol-abuse syndrome did not have an emphasis on repetitive drinking as the cause, whereas alcohol-dependence syndrome had an emphasis on repetitive drinking as well as dependence leading to illness. The mechanism of the changes in antioxidant levels in alcohol-abuse patients at the time of admission (onset) may be different with alcohol-dependent patients; for example, the former may be a manifestation of acute physiological stress, while the latter is a manifestation of long-term alcohol effects on the body. Third, there is possible heterogeneity in patient enrollment due to the alterations in the pathological standards. The inconsistency between the male and female ratios in some studies (particularly in some articles that studied only male patients) and different geographical regions may also contribute to the existence of publication bias. Therefore, we chose to use the combined effect size estimation method based on the results of the heterogeneity tests. In this case, the use of the RE model (*I*^2^ > 50%) might have amplified the publication bias of the small sample size study due to the application of the equal-weight method [135,136]. Although sensitivity analyses showed relatively good stability of the effect sizes of OS biomarkers, the small number of studies, the differences in the quality of individual samples, and the use of the RE model might also lead to the poor quality of results in this study. Additionally, the level of OS in female patients could not be studied due to the inclusion of articles focusing mostly on male patients. Future investigations should be conducted involving both genders at equal ratios to eliminate the possibility of gender bias.

In summary, to obtain in-depth pathological information about the altered levels of OS markers in AD patients, special attention should be given to the number of studies and sample sizes with statistical significance, excluding other confounding factors (e.g., smoking, diabetes, etc.) and designing experimental plans with scientific rigor, including age- and gender-matched controls, as well as other possible factors.

## 5. Conclusions

Scientific research on OS biomarkers is useful to provide an objective basis for clinical rehabilitation for AD subjects. Through the meta-analysis of published articles on the levels of OS in AD patients, we found that there may be a fatal level of OS in such patients, which might have caused serious health consequences. Amongst them, antioxidant enzymes such as SOD and GPx exhibited opposite trends in serum/plasma and RBC samples of AD patients, which could be used as the marker for characterizing OS in this subset of patients, especially in men. However, we should pay more attention to the changes in some critical antioxidant factors such as albumin, vitamin B, and homocysteine to prevent several chronic and irreversible degenerative diseases, including liver diseases, cardiovascular diseases, and metabolic syndromes, as well as neuropsychiatric disorders.

## Figures and Tables

**Figure 1 antioxidants-11-01919-f001:**
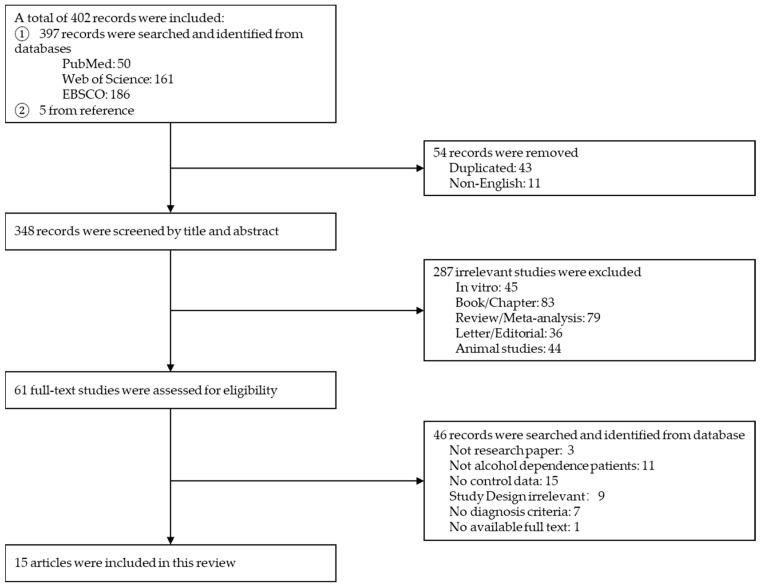
Process of searching and inclusion-exclusion.

**Figure 2 antioxidants-11-01919-f002:**
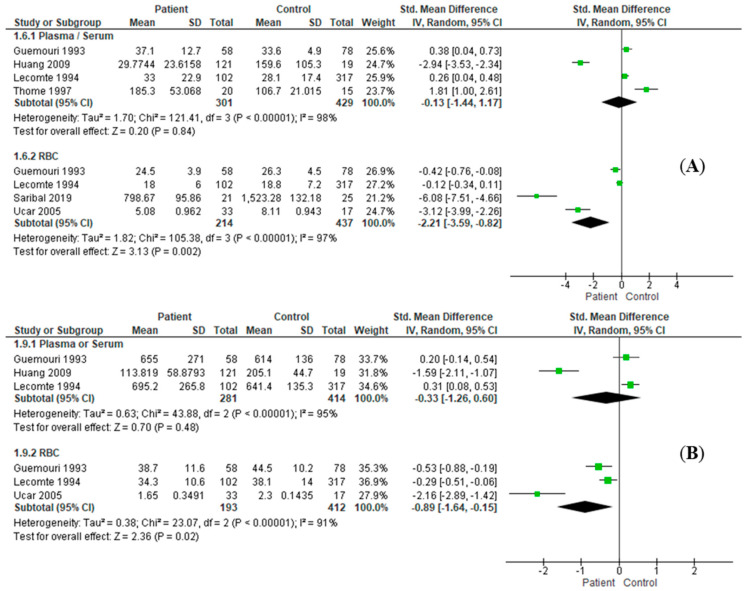
Comparing activities of antioxidative enzymes between alcohol-dependent patients and healthy controls in plasma/serum or RBC. Significantly different activities were found in SOD (**A**) in RBC (RE: SMD = −2.21, *I*^2^ = 97%, 95%CI = [−3.59, −0.82], *Z* = 3.13, *p =* 0.002) but not in plasma/serum, GPx (**B**) in RBC (RE: SMD = −0.89, *I*^2^ = 91%, 95%CI = [−1.64, −0.15], *Z* = 2.36, *p* = 0.02) but not in plasma/serum. CI, confidential interval; GPx, glutathione peroxidase; RBC, red blood cell; RE, random effect mode; SD, standard deviation; SMD, standard mean difference; SOD, superoxide dismutase.

**Figure 3 antioxidants-11-01919-f003:**
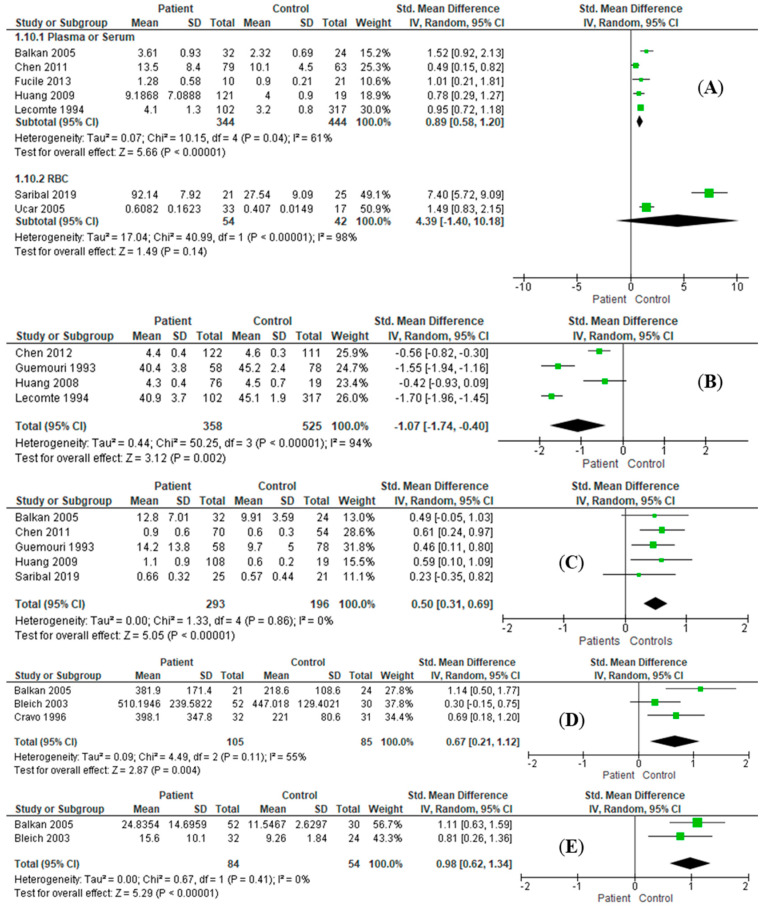
Comparing different levels of antioxidants between alcohol-dependent patients and healthy controls in plasma/serum or RBC. (**A**) Significant difference in MDA was found in plasma/serum (RE: SMD = 0.89, *I*^2^ = 61%, 95%CI = [0.58, 1.20], *Z* = 5.66, *p* < 0.001) but not in RBC. In plasma/serum, there were significant differences found in albumin (**B**) (RE: SMD = −1.07, *I*^2^ = 94%, 95%CI = [−1.74, −0.40], *Z* = 3.12, *p* = 0.002), bilirubin (**C**) (RE: SMD = 0.50, *I*^2^ = 0%, 95%CI = [0.31, 0.69], *Z* = 5.05, *p* < 0.001), B_12_ (**D**) (RE: SMD = 0.67, *I*^2^ = 55%, 95%CI = [0.21, 1.12], *Z* = 2.87, *p* = 0.004), and homocysteine (**E**) (RE: SMD = 0.98, *I*^2^ = 0%, 95%CI = [0.62, 1.34], *Z* = 2.81, *p* < 0.001). B_12_, vitamin B_12_; CI, confidential interval; MDA, malondialdehyde; RBC, red blood cell; RE, random effect mode; SD, standard deviation; SMD, standard mean difference.

**Figure 4 antioxidants-11-01919-f004:**
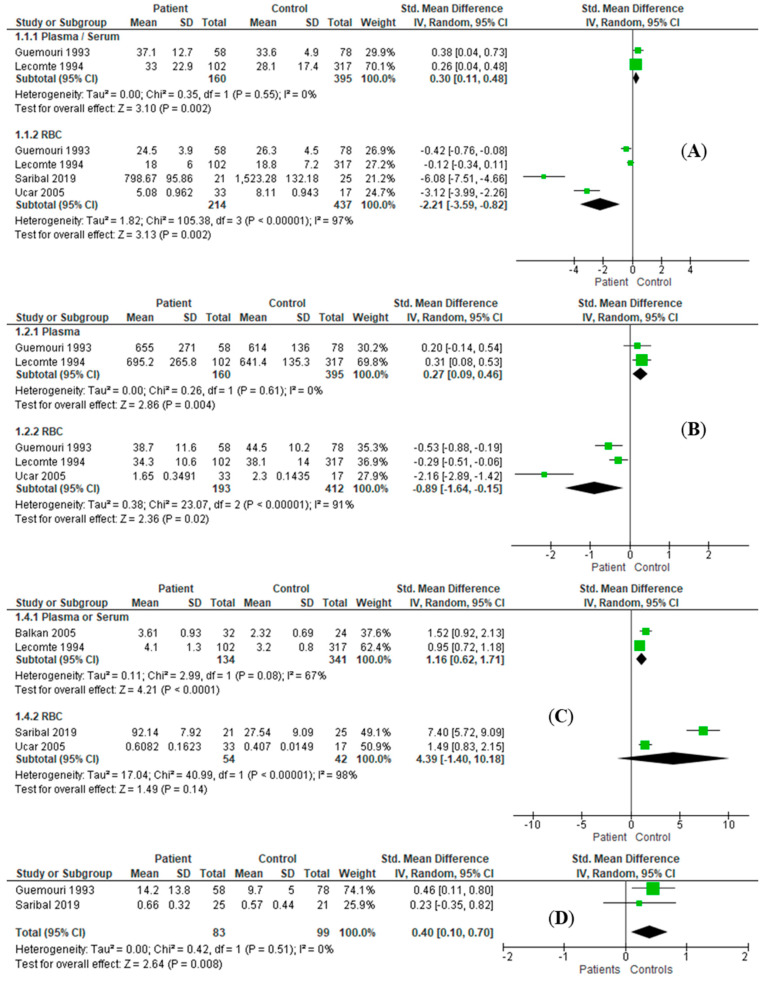
Different levels of antioxidants and pro-oxidants in plasma/serum or RBC between male alcohol-dependent patients and healthy male controls. (**A**) There are significant differences in SOD activity between male alcohol-dependent patients and healthy male controls in plasma/serum (RE: SMD = 0.30, I2 = 0%, 95%CI = [0.11, 0.48], Z = 3.10, *p* = 0.002) and RBC (RE: SMD = −2.21, I2 = 97%, 95%CI = [−3.59, −0.82], Z = 3.13, *p* = 0.002). (**B**) There are significant differences in GPx activity between male alcohol-dependent patients and healthy male controls in plasma/serum (RE: SMD = 0.27, I2 = 0%, 95%CI = [0.09, 0.46], Z = 2.86, *p* = 0.004) and RBC (RE: SMD = −0.89, I2 = 91%, 95%CI = [−1.64, −0.15], Z = 2.36, *p* = 0.02). (**C**) Significantly different level of MDA was found in plasma/serum (RE: SMD = 1.16, I2 = 67%, 95%CI = [0.62, 1.71], Z = 4.21, *p* < 0.001) but not in RBC. (**D**) Significantly different level of bilirubin was found in plasma/serum (RE: SMD = 0.40, I2 = 0%, 95%CI = [0.10, 0.70], Z = 2.64, *p* = 0.008). CI, confidential interval; GPx, glutathione peroxidase; MDA, malondialdehyde; RBC, red blood cell; RE, random effect mode; SD, standard deviation; SMD, standard mean difference; SOD, superoxide dismutase.

**Table 1 antioxidants-11-01919-t001:** Basic information of the included literature.

Study	Year	Patient		Control		Diagnosis Criteria	Drink History (Years)	Alcohol Consumption	Oxidative Index
		Size (M/F)	Age (Mn ± SD)	Size (M/F)	Age (Mn ± SD)				
Balkan	2005	32 (26/6)	48.6 ± 9.86 (40–60)	24 (18/6)	52.3 ± 10.2 (42–62)	DSM-IV	18.8 ± 9.04 (5–30)	225.9 ± 88.2 (g/day)	Plasma: bilirubin, MDA, diene conjugate, homocysteine, folic acid, vitamin B12
Bleich	2003	52 (34/18)	**M**: 45.59 ± 8.41 **F**: 48.44 ± 9.98	30 (16/14)	**M**: 48.34 ± 8.13 **F**: 48.00 ± 11.36	DSM-IV	**M**: 13.35 ± 5.74 **F**:12.61 ± 3.71	Lifetime drinking (kg) **M**: 1652.56 ± 1572.94 **F**: 685.44 ± 320.52	Plasma: total homocysteine, folic acid, B12, B6
Chen	2011	79 (67/12)	41 ± 7.0	63 (58/5)	40.7 ± 8.3	DSM-IV-IR	11.0 ± 7.5 (n = 75)	196.5 ± 105.0 (g/day)	Serum: MDA, total bilirubin, 8-OHdG
Chen	2012	124 (79/45)	40.0 ± 8.5	111 (73/38)	34.0 ± 9.8	DSM-IV-IR	11.6 ± 7.1 (n = 103)	Lifetime drinking (kg) 566.0 ± 484.0 (n = 103)	Serum: albumin
Cravo	1996	32 (24/8)	43(29–60)	31 (19/12)	36 (25–63)	DSM-III-R	≥5 years	**P**: 2.78 ± 1.32 (g/kg/day) **C**: ≤30 g/day	Serum: folate, pyridoxal phosphate, vitamin B12 RB**C**: Folate
Fucile	2013	10 (8/2)	45.7 ± 11.0	20 (16/4)	42.8 ± 8.4	DSM-IV-TR	NA	NA	Plasma: MDA
Guemouri	1993	58 (58/0)	42.1 ± 8.2 (26–61)	78 (78/0)	35.8 ± 5.7 (24–54)	DSM-III-R	13.7 ± 8.5	**P**: 197.1 ± 132.6 (g/day) **C**: 19.9 ± 20.6 (g/day, ≤88 g/d)	Plasma: bilirubin, albumin, SOD, GPx RB**C**: SOD, GPx, CAT
Huang	2008	76	41.2 ± 8.5	19	30.4 ± 10.4	DSM-IV-TR	12.4 ± 7.7	208.9 ± 100.7 (g/day)	Serum: SOD, MDA, total bilirubin, total albumin
Huang	2009	121 (113/8)	42.2 ± 9.0	19 (11/8)	30.4 ± 10.4	DSM-IV-TR	13.2 ± 8.5 (n = 113)	216.6 ± 107.1 (n = 114, g/day)	Serum: SOD, CAT, GSH, MDA, total bilirubin
Kapaki	2007	71 (58/13)	45 ± 11	61 (27/34)	44.8 ± 17.9	DSM-IV	16.7 ± 7.6	360 ± 258 (g/day)	Serum: protein carbonyl
Lecomte	1994	102 (102/0)	40.5 ± 8.8	317 (317/0)	36.2 ± 6.7	DSM-III-R	13.2 ± 8.3	**P**: 194.3 ± 140.7 (g/day) **C**: 10.6 ± 9.2 (g/day, ≤33 g/day)	Plasma: albumin, α-tocopherol, ascorbic acid, selenium, GPx, SOD, MDA RB**C**: GPx, SOD
Peng	2005	29 (28/1)	43.81 ± 10.41 (25–66)	19 (11/8)	30.33 ± 10.93 (21–57)	DSM-III-R	22.2 ± 10.5 (3–46)	271 ± 123.6 (120–660) (g/day)	Serum: total-bilirubin, total protein, albumin, uric acid, MDA, SOD, CAT, GR, GPX
Saribal	2019	21 (21/0)	28–52	25 (25/0)	28–52	DSM-IV-IR	NA	>80 (g/day)	RBC: SOD, CAT, GPx, MDA, Cu, Fe, Zn Serum: total bilirubin
Thome	1997	20 (18/2)	40 ± 8 (30–59)	15 (13/2)	33 ± 10 (23–57)	ICD-10 & DSM-III-R	14.3 ± 7.6 (4–35)	249 ± 99 (120–450) (g/day)	Serum: lactoferrin, SOD
Ucar	2005	33 (33/0)	I: 45.1 ± 6.3 (34–56) II: 41.6 ± 9.6 (27–62)	17 (17/0)	40.3 ± 8.4 (29–56)	ICD-10	I: 14.1 ± 6.9 II: 22.3 ± 9.7	NA	RBC (lysates and membranes): lipid peroxidation, GSH, GSSG, protein-bound GSH, GR, CAT, SOD, GPx

Note: 8-OHdG, 8-hydroxy-2′-deoxyguanosine; **C**, controls; CAT, catalase; Cu, cuprum; **F**, female; Fe, iron; GR, glutathione reductase; GPx, glutathione peroxidase; GSH, glutathione; GSSG, oxidized glutathione; **M**, male; MDA, malondialdehyde; Mn ± SD, mean ± standard deviation; **P**, patients; RBC, red blood cell; SOD, superoxide dismutase; Zn, zinc.

**Table 2 antioxidants-11-01919-t002:** Results of quality assessment by using the Newcastle–Ottawa scale for case-control studies.

Study	Selection				Comparability Control for Important Factor ^#^	Exposure			Score *
	Case Definition Is Adequate	Representativeness of the Cases	Controls Selection	Controls Definition		Ascertainment of Exposure	Same Method of Ascertain for Cases and Controls	Non-Response Rate	
Balkan 2005	★	★	★	★	★★	★	★	★	9
Bleich 2003	★	★	★	★	★★			★	7
Chen 2011	★	★	★	★	★★	★	★	★	9
Chen 2012	★	★	★	★	★★	★	★	★	9
Cravo 1996	★	★	★	★	★	★	★	★	8
Fucile 2013	★	★	★	★	★★	★	★	★	9
Guemouri 1993	★	★	★	★	★★	★	★	★	9
Huang 2008	★	★	★	★		★	★	★	7
Huang 2009	★	★	★	★		★	★	★	7
Kapaki 2007	★	★	★	★	★	★	★	★	8
Lecomte 1994	★	★	★	★	★★	★	★	★	9
Peng 2005	★	★	★	★	★	★	★	★	8
Saribal 2019	★	★	★	★	★	★	★	★	8
Thome 1997	★	★	★	★	★★	★	★	★	9
Ucar 2005	★	★	★	★	★	★	★	★	8

^#^ Comparability control of age and gender each gets a star, maximum of 2. * The maximum score is 9.

**Table 3 antioxidants-11-01919-t003:** The effect sizes of oxidative biomarker between alcohol-dependent patients and healthy controls.

Oxidative Stress Biomarkers		Alcohol-Dependent Patients vs. Healthy Controls	Patients vs. Controls in Male
SOD	P/S	RE: SMD = 0.13, I^2^ = 98%, 95%CI = [−1.44, 1.17], Z = 0.2 (*p* = 0.84)	RE: SMD = 0.30, I^2^ = 0%, 95%CI = [0.11, 0.48], Z = 3.10 (*p* = 0.002)
	RBC	RE: SMD = −2.21, I^2^ = 97%, 95%CI = [−3.59, −0.82], Z = 3.13 (*p* = 0.002)	RE: SMD = −2.21, I^2^ = 97%, 95%CI = [−3.59, −0.82], Z = 3.13 (*p* = 0.002)
CAT	RBC	RE: SMD = −2.68, I^2^ = 98%, 95%CI = [−6.21, 0.84], Z = 1.49 (*p* = 0.14)	RE: SMD = −2.58, I^2^ = 98%, 95%CI = [−6.00, 0.84], Z = 1.48 (*p* = 0.14)
GPx	P/S	RE: SMD = −0.33, I^2^ = 95%, 95%CI = [−1.26, 0.60], Z = 0.70 (*p* = 0.48)	RE: SMD = 0.27, I^2^ = 0%, 95%CI = [0.09, 0.46], Z = 2.86 (*p* = 0.004)
	RBC	RE: SMD = −0.89, I^2^ = 91%, 95%CI = [−1.64, −0.15], Z = 2.36 (*p* = 0.02)	RE: SMD = −0.89, I^2^ = 91%, 95%CI = [−1.64, −0.15], Z = 2.36 (*p* = 0.02)
GSH	RBC	RE: SMD = 0.15, I^2^ = 98%, 95%CI = [−3.05, 3.35], Z = 0.09 (*p* = 0.93)	
MDA	P/S	RE: SMD = 0.89, I^2^ = 61%, 95%CI = [0.58, 1.20], Z = 5.66 (*p* < 0.001)	RE: SMD = 1.16, I^2^ = 67%, 95%CI = [0.62, 1.71], Z = 4.21 (*p* < 0.001)
	RBC	RE: SMD = 4.39, I^2^ = 98%, 95%CI = [−1.40, 10.18], Z = 1.49 (*p* = 0.14)	RE: SMD = 4.39, I^2^ = 98%, 95%CI = [−1.40, 10.18], Z = 1.49 (*p* = 0.14)
Albumin	P/S	RE: SMD = −1.07, I^2^ = 94%, 95%CI = [−1.74, −0.40], Z = 3.12 (*p* = 0.002)	RE: SMD = −3.04, I^2^ = 99%, 95%CI = [−6.39, 0.31], Z = 1.78 (*p* = 0.08)
Bilirubin	P/S	RE: SMD = 0.50, I^2^ = 0%, 95%CI = [0.31, 0.69], Z = 5.05 (*p* < 0.001)	RE: SMD = 0.40, I^2^ = 0%, 95%CI = [0.10, 0.70], Z = 2.64 (*p* = 0.008)
B6	P/S	RE: SMD = −0.86, I^2^ = 91%, 95%CI = [−2.07, 0.35], Z = 1.39 (*p* = 0.16)	
B12	P/S	RE: SMD = 0.67, I^2^ = 55%, 95%CI = [0.21, 1.12], Z = 2.87 (*p* = 0.004)	
Folic Acid	P/S	RE: SMD = −0.45, I^2^ = 94%, 95%CI = [−1.63, 0.74], Z = 0.74 (*p* = 0.46)	
Homocysteine	P/S	RE: SMD = 0.98, I^2^ = 0%, 95%CI = [0.62, 1.34], Z = 2.81 (*p* < 0.001)	

B6, vitamin B6; B12, vitamin B12; CAT, catalase; CI, confidence interval; GPx, glutathione peroxidase; GSH, glutathione; MDA, malondialdehyde; P/S, plasma or serum; RBC, red blood cell; RE, random model; SMD, standardized mean difference; SOD, superoxide dismutase.

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
