# Peer review of "The Status of Oxidative Stress in Patients with Alcohol Dependence: A Meta-Analysis"

_antioxidants, 2022, doi:10.3390/antiox11101919_

Round 1
Reviewer 1 Report
In this paper, the authors analyze the relationship between alcohol-induced oxidative stress (OS) and alcohol dependence (AD) using a meta-analysis method.
The authors performed a detailed analysis of various biomarkers with appropriate methods, and the results of this paper provide us with very important information.
Naturally, it was expected that not only changes in various antioxidant enzymes but also the subject's nutritional status, including vitamin levels, would have a significant impact, and this paper objectively proves this prediction.
It has already been reported that the SOD activity in the blood of males is elevated in oxidative stress-related diseases, which is consistent with the authors' results.
Based on the above, we believe that this paper is worthy enough to be published in this journal.
Author Response
Dear reviewer,
Please see the response attachment.
Best regards!

Reviewer 2 Report
The meta-analyze paper on the status of oxidative stress in patients with alcohol dependence excitinging research in this field where there are many papers with different conclusions that do not allow helpful conclusions for pathophysiology and clinical practice.
But I think that it is necessary to review:
Line 267: Really, the cytoglobin has the same SOD-activity as copper/zinc-SOD (SOD1) or manganese-SOD (SOD2), but at this point of discussion,n it is not necessary, and may even mislead, to mention two isoenzymes because there is no Mn-SOD or mitochondrial in erythrocytes.
Line 274 -284: The plasma/serum SOD activity is too low, and the measuring methodology could be essential.
368-370. Albumin is also a protein with antioxidant properties in plasma/serum ( FEBS Letters Volume 582, Issue 13, 11 June 2008, Pages 1783. Still, theut the authors neither describe it nor discuss the possible opposing results between the lower level in serum/plasma and high level in SOD activity serum/plasma.
359. The authors may include this reference when discussing the usefulness of Vit B12 in hepatic toxicity Pol Merkur Lekarski. . 2010 Feb;28(164):122-5.
Author Response
Dear Reviewer,
Please see the response attachment,
Best regards!

Reviewer 3 Report
The status of oxidative stress in patients with alcohol dependence: a meta-analysis
This article provides a systematic review of 15 publications into alcohol dependence. Overall, the article seems clear, well executed and written, but there are a few minor points, detailed below, that need addressing. In addition to these there is a major concern that I have concerning the completeness of this work. The authors highlight in their limitations section the fact that over the period of the studies they have included, that both DSM and ICD have changed their diagnostic criteria of alcohol related disorders considerably. This has a concerning implication on the inclusion criteria they have used in their article selection. In DSM-5 the previous diagnoses of alcohol abuse and alcohol dependence have now been combined into alcohol use disorder. This implies that just searching the databases for alcohol dependence will miss many possible studies where they have used the current DSM/ICD diagnosis of alcohol use disorder… For example, a quick search of pubmed for just the key words ‘alcohol dependence’ and ‘alcohol use disorder’ produces 105,503 and 115,130 hits respectively, an extra ~10,000 extra studies that may have included other inclusion criteria, increasing the overall pool for of studies for this, otherwise good, meta analysis. Especially as another limitation was the ‘lack’ of appropriate studies, 15. The authors must address this choice of search terms in their response to this and in the manuscript, indicating why they chose not to use alcohol use disorder and include the limitations of this choice on the results and conclusions from their current analysis.
Minor points
Page 2 line 48 – the authors say “…inducing a bunch…”, this is a rather colloquial phrase and perhaps the authors should change it to …inducing a number or many…
Page 2 line 63 – “Therefore, alcohol and its toxic metabolites are the sole cause…” This is a very strong statement. Are the authors suggesting that there is NO other cause of ROS/RNS etc? I find this conclusion rather hard to accept and I’m not convinced this is the conclusion that ref 23 draws... This statement needs moderation.
Page 4 line 118 – The authors have abbreviated Newcastle-Ottawa scale as NOS. This abbreviation has already been used for “nitric oxide synthase”. You can’t used the same abbreviation it is confusing, please change one of these.
Page 5 lines 149-151 – the authors should delete the ‘instructions to authors’…
Page 7 and 13 line 167 and 262 respectively – It is ‘compare(d) with’ not ‘compare(d) to’, please change.
Page 16 line 406 – the authors state, “…we found that there was a fatal level of OS in such patients…”. I think this statement is also rather strong. Were these ‘patients’ dead? As this is the assumption to be made from the word fatal… I think the authors should revise this statement.
Author Response
Dear reviewer,
Please see the response attachment,
Best regards!

Round 2
Reviewer 2 Report
Accept
Reviewer 3 Report
You have addressed the comments as required.